# Impaired Innate Immunity in Pediatric Patients Type 1 Diabetes—Focus on Toll-like Receptors Expression

**DOI:** 10.3390/ijms222212135

**Published:** 2021-11-09

**Authors:** Katarzyna Kurianowicz, Maria Klatka, Agnieszka Polak, Anna Hymos, Dominika Bębnowska, Martyna Podgajna, Rafał Hrynkiewicz, Olga Sierawska, Paulina Niedźwiedzka-Rystwej

**Affiliations:** 1Department of Pediatric Endocrinology and Diabetology, Medical University of Lublin, Gębali 1 St., 20-093 Lublin, Poland; katarzyna.kozyra88@gmail.com (K.K.); mariaklatka@wp.pl (M.K.); 2Department of Endocrinology, Medical University of Lublin, Jaczewskiego 8 St., 20-954 Lublin, Poland; agnieszka.polak@icloud.com; 3Department of Experimental Immunology, Medical University of Lublin, Chodźki 4a St., 20-093 Lublin, Poland; annahymos@gmail.com; 4Institute of Biology, University of Szczecin, 71-412 Szczecin, Poland; dominika.bebnowska@usz.edu.pl (D.B.); rafal.hrynkiewicz@usz.edu.pl (R.H.); olga.sierawska@gmail.com (O.S.); 5Department of Clinica Immunology and Immunotherapy, Medical University of Lublin, Chodźki 4a St., 20-093 Lublin, Poland; marpodgajna@gmail.com

**Keywords:** type 1 diabetes mellitus, children, Toll-like receptors

## Abstract

Type 1 diabetes (DM1) is classified as an autoimmune disease. An uncontrolled response of B and T lymphocytes to the body’s own tissues develops in the absence of immune tolerance. The main aim of the study was to evaluate the effect of the duration of type 1 diabetes in children on the expression of TLR receptors and the relationship with the parameters of glycemic control in patients. As a result, we showed significant differences in the level of TLR2, TLR4 and TLR9 expression in patients with DM1 in the early stage of the disease and treated chronically compared to the healthy group. Additionally, in this study, we found that the numbers of CD19+ B cells, CD3+ CD4+, CD3+ CD8+ T cells and NK cells are different for newly diagnosed DM1 individuals, patients receiving chronic treatment and for healthy controls, indicating an important role of these cells in killing pancreatic beta cells. Moreover, higher levels of IL-10 in patients with newly diagnosed DM1 have also been found, confirming the reports found in the literature.

## 1. Introduction

Diabetes mellitus is one of the most common chronic metabolic diseases. The United Nations recognized that it is the first and only non-communicable disease to be the epidemic of the 21st century [1]. The term “diabetes” as defined by the American Diabetes Association (ADA) is a group of metabolic diseases characterized by chronic hyperglycemia resulting from disturbances in insulin secretion and/or function. Insufficient insulin secretion and/or reduced tissue response to insulin impairs its action in target tissues, which affects the metabolism of carbohydrates, lipids, and proteins [2].

Type 1 diabetes mellitus (DM 1) is one of the most common chronic metabolic diseases in the population under the age of 18. According to the 2017 report by the International Diabetes Federation (IDF), the number of children up to 14 years old diagnosed with type 1 diabetes has reached almost 586,000 [3]. Incidence reaches 3% of the child population. It has been estimated that approximately 96,000 new cases of diabetes are revealed annually [3,4]. Type 1 diabetes is caused by the destruction of the β-cells of the pancreas, which leads to an absolute insulin deficiency. The principles of diagnosing diabetes are based on the assessment of blood glucose levels and the presence of characteristic clinical symptoms [2,5]. Common symptoms of diabetes include polyuria, increased thirst (polydipsia) and weight loss. This triad may be associated with bedwetting (nocturia), involuntary urination, weakness, behavioral changes, candidiasis in the external urinary tract, increased appetite, and blurred vision. Hyperglycaemia leads to ketoacidosis or, less commonly, non-keto hyperosmolar syndrome, followed by stupor or coma. Untreated diabetes leads to death [5]. Type 1 diabetes mellitus is a multifactorial disease, but its etiopathogenesis is not fully understood. According to the currently adopted model of the pathogenesis of type 1 diabetes, proposed in 1986 by George Eisenbarth, the development of the disease requires the interaction of three components: genetic predisposition, environmental factors and the immune system [6,7]. Considering the prevalence of diabetes and its serious consequences, we believe that any research that brings us closer to increasing our knowledge of each of these aspects can be very valuable both for improving the treatment and quality of life of patients.

The autoimmune disorders in DM1 are triggered by an impaired immune response. Toll-like receptors (TLRs) are receptors classified as pattern recognition receptors (PRR). TLR receptors combine non-specific immunity with specific immunity. By recognizing molecular patterns associated with pathogens, they initiate a non-specific immune response. Moreover, they enable cells of the immune system to distinguish self and foreign antigens [8]. TLRs are expressed particularly on the surface of antigen-presenting cells—dendritic cells, macrophages, B lymphocytes, and also on the surface of CD4+, CD8+ T cells and CD4+ CD25+ FoxP3+ regulatory T cells [9,10]. TLRs play a dual role in modulating the immune response. On the one hand, they enhance innate and adaptive defense mechanisms against pathogens, while on the other hand, they are involved in the autoimmune destruction of β-cells in DM1 by T cells. It has been suggested that manipulating TLR signaling may protect against DM1 [11]. CD4+ CD25+ FoxP3+ (Treg) T cells play a major role in regulating the inflammatory response within the pancreatic islets [12]. Their role is to maintain the balance between inflammatory cytokines and tolerance to own tissues. Reducing the proportion of these populations or disrupting their function is one of the key factors in the pathogenesis of type 1 diabetes [13,14,15,16].

Activation of appropriate signaling pathways involving TLR receptors in predisposed individuals may reduce the likelihood of developing diabetes in the future. Assessing TLRs can be the key in preventing uncontrolled inflammation and reducing disease progression. The main aim of this study was to assess the effect of the duration of type 1 diabetes in children on the expression of TLR receptors and the relationship with the parameters of glycemic control in patients. Additionally, we assessed the concentrations of IFN-γ, IL-2, IL-4 and IL-10 in the plasma of individuals with type 1 diabetes and healthy group.

## 2. Results

The characteristics of the study and the control Group Ire presented in Table 1. Group I—patients with newly diagnosed type 1 diabetes (<3 months from diagnosis), group II—patients with chronic type 1 diabetes (>3 months from diagnosis).

### 2.1. Assessment of the Subpopulation of Peripheral Blood Lymphocytes in the Study and Control Group

The percentage and absolute values of CD3+ T cells in the peripheral blood of patients with type 1 diabetes and in the control group were analyzed. The results are presented in Table 1.

The conducted analysis showed a significantly higher percentage (*p* = 0.009) and a higher number (*p* = 0.036) of CD3+ T cells in the group of children with type 1 diabetes compared to the control group. There was a significantly higher percentage (*p* = 0.016) and higher number (*p* = 0.019) of CD3+ T cells in the newly diagnosed diabetic group (<3 months from diagnosis) than in the control group. It was also found that the percentage of CD3+ T cells in the peripheral blood of patients suffering from type 1 diabetes for more than 3 months was higher than the percentage of these cells in the control group (*p* = 0.042).

The results showing the percentage and absolute values of CD19+ B cells in the peripheral blood of patients with type 1 diabetes and in the control group are provided in Table 1.

The conducted analysis showed a significantly higher number (*p* = 0.015) of CD19+ B lymphocytes in the group of children with type 1 diabetes than in the control group. There was a significantly higher percentage (*p* = 0.016) and higher number (*p* = 0.003) of CD19+ B lymphocytes in the group of patients with newly diagnosed diabetes (<3 months from diagnosis) than in the control group. There was a higher percentage (*p* = 0.019) and a higher number (*p* = 0.019) of CD19+ B cells in children with type 1 diabetes for less than 3 months compared to chronic DM1 patients.

The percentage and absolute values of CD3+ CD4+ T cells in the peripheral blood of patients with type 1 diabetes and in the control Group Ire presented in Table 1.

There was a statistically significantly higher percentage (*p* = 0.038) and a higher number (*p* = 0.028) of CD3+ CD4+ T cells in the group with type 1 diabetes than in the control group. There was a significantly higher number (*p* = 0.022) of CD3+ CD4+ T cells in the group of patients with newly diagnosed diabetes (<3 months from diagnosis) than in the control group. 

The percentage and absolute values of CD3+ CD8+ T cells in patients with type 1 diabetes and in the control group are presented in Table 1.

The analysis showed a significantly higher number (*p* = 0.038) of CD3+ CD8+ T cells in the group of children with type 1 diabetes than in healthy subjects. There was a significantly higher percentage (*p* = 0.015) and higher number (*p* = 0.015) of CD3+ CD8+ T cells in the group of patients with newly diagnosed type 1 diabetes than in the control group.

The assessment of the relationship between the ratio of CD3+ CD4+ lymphocytes to CD3+ CD8+ in the studied groups did not show any statistically significant differences (Table 1).

The absolute values and the percentage of NK cells in children with type 1 diabetes and in the control group are presented in Table 1.

The percentage of NK cells in the DM1 group was statistically significantly lower than in the control group (*p* = 0.007).

There was a significantly lower percentage (*p* < 0.001) and a lower number (*p* = 0.035) of NK cells in the group with DM1 lasting less than 3 months than in the control group. There was also an increase in the percentage (*p* = 0.001) and the number (*p* = 0.041) of NK cells in patients with long-term diabetes mellitus type 1 compared to the group with newly diagnosed DM1.

### 2.2. Assessment of B CD19+, T CD4+ and T CD8+ Lymphocytes with TLR2, TLR4, TLR9 Expression in the Study and Control Groups

The detailed results of the assessment of B CD19+, T CD4+ and T CD8+ lymphocytes with TLR2, TLR4, TLR9 expression in the study and control groups with statistical analysis are presented in Appendix A.

There were no statistically significant differences between the study group and the control group, as well as between both study groups in the percentage and the number of CD19+ B cells with TLR2 expression in the peripheral blood.

It was shown that the percentage (*p* = 0.018) and the number (*p* = 0.003) of lymphocytes with the CD4+ TLR2+ phenotype in the peripheral blood of patients with type 1 diabetes mellitus were higher than in the control group. The percentage (*p* = 0.010) and number (*p* = 0.002) of CD4+ TLR2+ T cells in patients with newly diagnosed type 1 diabetes mellitus were higher compared to the control group. The number (*p* = 0.040) of CD4+ TLR2+ T cells in children with long-term type 1 diabetes mellitus was higher than in the control group (Figure 1A).

The conducted analysis showed that the percentage of CD8+ TLR2+ lymphocytes in the peripheral blood of children with type 1 diabetes was lower than in the control group (*p* = 0.026). The percentage of CD8+ TLR2+ lymphocytes in the group with DM1 < 3 months was statistically significantly lower than in the control group (*p* = 0.025). Those differences were not visible when we compared the absolute values of those cells (Figure 1B).

In children with type 1 diabetes for more than 3 months, the percentage (*p* = 0.021) and number (*p* = 0.018) of CD19+ TLR4+ B cells were higher than in healthy subjects (Figure 1C).

The number of lymphocytes with the CD4+ TLR4+ phenotype in the peripheral blood of children with chronic DM1 patients was higher than in the control group (*p* = 0.027) (Figure 1D).

It was shown that the number of cells with the CD8+ TLR4+ phenotype in the peripheral blood of children suffering from DM1 compared to the number of these cells in the control group was statistically significantly higher (*p* = 0.040). It was found that the percentage (*p* = 0.006) and the number (*p* = 0.004) of CD8+ TLR4+ T cells in the peripheral blood of patients with DM1 for more than 3 months compared to the percentage of these cells in the control group were statistically significantly higher (Figure 1F). It was also found that the percentage of CD8+ TLR4+ T lymphocytes in the group of children suffering from DM1 below 3 months compared to children suffering from DM1 for more than 3 months was statistically significantly lower (*p* = 0.035).

The percentage of CD19+ B cells expressing TLR9 in the group with newly diagnosed type 1 diabetes mellitus compared to the control group was significantly lower (*p* = 0.004). The percentage (*p* = 0.001) and the number (*p* = 0.003) of CD19+ TLR9+ B cells in the group of children suffering from DM1 below 3 months were statistically significantly lower than in children suffering from DM1 for over 3 months (Figure 1F).

There were no statistically significant differences in the percentage and number of CD4+ T cells with TLR9 expression between the studied groups (Figure 1G).

It was found that the percentage of CD8+ TLR9+ T cells in children with type 1 diabetes less than 3 months was lower compared to the control group (*p* = 0.040). The percentage of CD8+ TLR9+ T cells in the group of children with newly diagnosed type 1 diabetes mellitus was lower than in children with long-term type 1 diabetes (*p* = 0.016). Those differences did not show the statistical power in absolute values (Figure 1H).

### 2.3. Assessment of the Concentration of IFN-γ, IL-2, IL-4, and IL-10 in the Peripheral Blood Plasma in the Study and Control Groups

The detailed results of the assessment of the concentration of IFN-γ, IL-2, IL-4, and IL-10 in the peripheral blood plasma in the study and control groups with statistical analysis are presented in Appendix A.

There were no statistically significant differences in IFN-γ concentrations between the group of subjects with type 1 diabetes and healthy subjects. There were also no statistically significant differences in the concentration of IFN-γ in the peripheral blood plasma in the group of children with type 1 diabetes with different duration of the disease (Figure 2A).

It was found that the concentration of IL-2 in children with long duration type 1 diabetes was higher than in healthy subjects (*p* = 0.030, Figure 2B).

There were no statistically significant differences between the study group and the control group, as well as between patients with type 1 diabetes of various duration in the concentration of IL-4 (Figure 2C).

The analysis showed that the concentration of IL-10 in the peripheral blood plasma in children suffering from DM1 below 3 months, compared to the concentration of these cytokines in the control group, was statistically significantly higher (*p* = 0.005). The concentration of IL-10 in the peripheral blood plasma of patients suffering from DM1 for less than 3 months compared to patients suffering from DM1 more than 3 months was higher (*p* = 0.003) (Figure 2D).

### 2.4. Assessment of Interdependencies between the Investigated Immunological and Clinical Parameters in the Study Group

The performed analysis showed a negative correlation of the percentage (R = −0.78; *p* < 0.05) (Figure 3) and the number (R = −0.69, *p* < 0.05) of CD19+ TLR9+ B cells with the level of glycated hemoglobin in a group of patients with newly diagnosed type 1 diabetes.

Statistically significant relationships were also found between the percentage (R = −0.67; *p* < 0.05) (Figure 4) and the number (R = −0.53, *p* < 0.05) of CD19+ TLR9+ B cells and the level of fructosamine in a group of patients with newly diagnosed type 1 diabetes.

There was a negative correlation between the percentage (R = −0.83, *p* < 0.05) (Figure 5) and the number (R = 0.67, *p* < 0.05) of CD8+ TLR9+ T cells with the level of glycosylated hemoglobin in people with DM1 of short duration (<3 months from diagnosis).

There was also a relationship between the percentage (R = −0.63, *p* < 0.05) (Figure 6) and the number (R = −0.51, *p* < 0.05) of CD8+ TLR9+ T cells and the level of fructosamine in the tested group (<3 months from diagnosis).

### 2.5. Assessment of the Relationships between B CD19+, T CD4+, and T CD8+ Lymphocytes with TLR2, TLR4, TLR9 Expression, and Duration Time after the Onset of DM1

The performed analysis showed a correlation between CD19+ TLR9+ B cells (R = 0.3091, *p* < 0.05) and duration time after the onset of DM1 (Figure 7).

There was also a relationship between CD8+ TLR4+ T cells (R = 0.3081, *p* < 0.05) and duration time after the onset of DM1 (Figure 8).

Statistically significant relationships were also found between CD8+ TLR9+ T cells (R = 0.2827, *p* < 0.05) and duration time after the onset of DM1 (Figure 9).

## 3. Discussion

The uncontrolled response of B and T lymphocytes towards the body’s own tissues develops in the absence of immune tolerance. The mechanisms responsible for the autoimmunological process include the activation of antigen-presenting cells, which promote the enhancement of autoantigen-presentation, and molecular mimicry [17]. It has been suggested that TLRs may also play an important role in the development of autoimmune diseases [18]. Inappropriate activation of receptors by endogenous and exogenous ligands and the influence of TLR-dependent pathways on Treg function contribute to the development of these diseases [17].

In this study, the percentage and number of selected lymphocytes expressing TLR2, TLR4, and TLR9 receptors in children with type 1 diabetes mellitus were assessed. The study group was divided into patients with newly diagnosed DM1, i.e., those who were diagnosed in previous 3 months and in patients suffering from chronic diabetes (>3 months). The conducted research showed significant differences in both groups and in comparison with the control group. A significantly lower percentage of CD19+ B lymphocytes (*p* = 0.004) and CD8+ T lymphocytes (*p* = 0.040) with TLR9 expression were found in the group of patients with newly diagnosed diabetes as compared to the control group. The obtained results may confirm the reports of Nündel et al. [19] on the participation of the TLR9 receptor in autoimmune diseases. There was a higher percentage of CD4+ TLR2+ lymphocytes (*p* = 0.010) and a significantly lower percentage of CD8+ TLR2+ lymphocytes (*p* = 0.025). In the group of patients treated for more than 3 months, a higher percentage of CD19+ TLR4+ lymphocytes (*p* = 0.021) and CD8+ TLR4+ T cells (*p* = 0.006) and a higher number of CD4+ TLR2+ T cells (*p* = 0.040) and CD4+ TLR4+ were found. (*p* = 0.027). The presented results are consistent with the reports of Devaraj et al. [20,21], in which a much higher expression of these receptors was found, and an increase in their ligands was observed.

A comparative assessment of the percentage and number of lymphocytes expressing TLR receptors between the groups of patients was also performed. A significantly lower percentage of CD19+ TLR9+ B cells (*p* <0.001) and CD8+ TLR9+ T cells (*p* = 0.016) was demonstrated in patients with newly diagnosed diabetes. In the group of subjects with type 1 diabetes (<3 months from diagnosis), a negative correlation was found between the percentage and the number of B lymphocytes with the CD19+ TLR9+ phenotype, as well as the percentage and the number of T lymphocytes with the CD8+ TLR9+ phenotype with HbA1c and the concentration of fructosamine. The results of the research suggest a change in the percentage of CD19+ B cells and CD8+ T cells with TLR9 expression depending on the time since the diagnosis of DM1. It was found that there is a relationship between the percentage and number of CD19+ B cells and CD8+ T cells expressing TLR9 and the parameters of glycemic control. The obtained results are consistent with the reports of Gupta et al. [22], who showed a correlation between the reduction of TLR9 expression, and the abnormal glucose concentration found in diabetic patients. Moreover, additional analysis of the obtained results showed that in the case of CD19+ TLR9+ B cells and CD8+ TLR4+ and CD8+ TLR9+ T cells, their values correlated statistically with the duration of the disease.

The pathogenesis of type 1 diabetes includes the abnormal activation of many cells: B cells, macrophages, dendritic cells, CD4+ T cells, and CD8+ T cells [23]. After stimulation with autoantigens, CD4+ T cells are activated, which differentiate into T helper cells (Th1, Th2 and Th17). IL-12, which is released by antigen-presenting cells, induces the production of IFN-γ and IL-2 by Th1 lymphocytes. The concentration of these cytokines correlates with the pathogenesis of type 1 diabetes [24]. IL-2 and IFN-γ activate CD8+ T cells to differentiate into effector CD8+ cytotoxic T lymphocytes (CTLs). CTLs release granzymes and perforins that are toxic to pancreatic beta cells [25]. CD8+ T cells are believed to be involved in the destruction of pancreatic beta cells. After 5 years of type 1 diabetes mellitus, a decrease in these lymphocytes and a correlation with glucose concentration and inflammation have been demonstrated [26,27]. Sarikonda et al. [28] showed that activation of CD4+ T cells is characteristic of both the pathogenesis of type 1 and type 2 diabetes, while activation of CD8+ T cells is unique in type 1 diabetes and leads to massive destruction of pancreatic beta cells.

The present study showed a significantly higher number of CD3+ T cells, CD19+ B cells, CD3+ CD4+ T cells and CD3+ CD8+ T cells in the study group (<3 months from diagnosis) than in the control group. The obtained results confirm the involvement of lymphocytes at the early stage of the disease and are consistent with the reports of Sarikonda et al. [28].

CD4+ CD25+ T cells play an important role in preventing autoimmunity. These lymphocytes are functionally regulated by signaling mediated by TLRs. Filippi et al. [29] showed that treating mice with TLR2 agonists before developing diabetes reduced the likelihood of developing type 1 diabetes by increasing the number of CD4+ CD25+ T cells and imparting tolerogenic properties to DC. It was then demonstrated that TLR2 signaling improves immunoregulation and prevents type 1 diabetes.

NK cells that produce IFN-γ contribute to the activation of autoreactive CD4+ and CD8+ T cells [30]. The presence of the NKp46 receptor is critical for the development of diabetes mellitus. Gur et al. [31] showed that the use of a protein fused with this receptor in mice reduced the ability to kill pancreatic beta cells and inhibited the development of diabetes mellitus. Alba et al. found that NK depletion also had a similar effect [32]. Similarly, the depletion of Treg cells leads to the accumulation of NK cells in the environment of the pancreas [30]. NKp46 ligand expression has also been found on pancreatic beta cells [33]. Gur et al. [31] confirmed that immunization with anti-NKp46 antibodies lowered diabetic incidents in mice. Zhang et al. [34] demonstrated a reduction in the percentage of NK cells in children with type 1 diabetes. Impaired ability of NK cells to produce IFN-γ was also detected after stimulation with phorbol myristate acetate (PMA) and ionomycin [34]. Studies have shown [34] that lowering the percentage of NK cells is involved in the pathogenesis of diabetes. It has been suggested that a reduction in the ability of NK cells to secrete cytokines is associated with viral factors that impair NK cell function [31].

Our own research showed a significantly lower percentage (*p* < 0.001) and number (*p* = 0.035) of NK cells in the study group (<3 months from diagnosis) than in the control group. There was also an increase in the percentage (*p* = 0.001) and the number (*p* = 0.041) of these cells in chronically ill patients compared to the group with newly diagnosed diabetes. The obtained results are consistent with the reports of Zhang et al. [34] and may suggest the involvement of these cells in the pathogenesis of the disease.

Binding of PAMPs by the TLR receptor induces the maturation of antigen-presenting cells and the production of pro-inflammatory cytokines and chemokines [35]. Alkanani et al. [36] assessed TLR—dependent release of IL-6 and IL-1β by monocytes and dendritic cells in patients susceptible to the development of type 1 diabetes. An imbalance between pro-inflammatory and regulatory mechanisms has been demonstrated, which in patients predisposed to the development of diabetes may lead to the development of full-blown diabetes in the future [36].

Zhang et al. [34] assessed cytokine levels in patients with type 1 diabetes, which showed a significantly higher concentration of IFN-γ and IL-2 compared to the control group. Rydén et al. [37] found a reduced percentage of Th2 lymphocytes in people at high risk of developing diabetes. It has been suggested that inducing a Th2-mediated response towards autoantigens in these individuals could keep the C-peptide at a residual level. Consequently, lowering the activation of Th1 lymphocytes may contribute to decreased risk of pancreatic beta cell destruction [37]. Studies by He et al. [38] have shown significantly higher levels of IL-1α, tumor necrosis factor (TNF-alpha), IL-6 and IL-12 in children with type 1 diabetes [38]. The obtained results suggested that there is a dysfunction of the regulatory mechanisms, including disturbances in the balance of Th1 and Th2 lymphocytes. In children with type 1 diabetes, excessive activation of the cytokines secretion by Th1 lymphocytes is observed. At the same time, the lack of Th2 lymphocyte response contributes to the disturbance of homeostasis, which leads to pancreatic cell damage and disease progression [38]. Ligand binding by TLR leads to the stimulation of Th1 cytokine production [39]. Mortezagholi et al. [40] showed that in SLE patients, TLR9 expression on the surface of B lymphocytes is significantly higher and correlates with the production of anti-dsDNA antibodies. TLR9 mediates the production of IFN-α, which is activated by autoantibody complexes [40].

He et al. [38] demonstrated much higher concentration of IL-10 in the blood plasma, which exhibits immunosuppressive properties. On the opposite, Kikodze et al. [41] observed a much lower concentration of IL-10 in the study group [41]. Esposito et al. [42] conducted a study in which they assessed the concentration of IL-10 in obese patients. The authors showed significantly higher levels of this cytokine in these patients, along with an increase in plasma levels of IL-6 and CRP, than in non-obese patients. The evaluation of IL-10 in patients with obesity and associated lipid metabolism disorders and arterial hypertension showed a reduction in IL-10 concentration compared to obese subjects who did not show any concomitant abnormalities. IL-10 lowers the production of pro-inflammatory cytokines, and the observed higher concentration of IL-10 may be an attempt to inhibit the increasing production of pro-inflammatory cytokines. On the other hand, a reduction of IL-10 production may indicate an impairment of this pathway. It can be assumed that a decrease in IL-10 concentration may signal the exhaustion of the mechanisms of physiological metabolic control [42]. Measurement of IL-10 concentration may indicate the pro-inflammatory cytokines activation [43]. Pestana et al. [44] demonstrated much higher concentrations of IL-6 and TNF-α along with a higher concentration of IL-10 in the blood plasma of patients with type 1 diabetes. Our own research showed a significantly higher concentration of IL-10 in the study group (<3 months from diagnosis) than in the control group (*p* = 0.005) and in the group of chronic DM1 patients (*p* = 0.024). The results of the assessment of IL-10 concentration from our own studies may confirm the suggestion that although IL-10 is an anti-inflammatory cytokine, its increase may suggest a compensatory effect, counterbalancing the increase in the concentration of pro-inflammatory cytokines.

Another cytokine that may participate in the pathogenesis of type 1 diabetes in children is IL-2. Studies by Dogan et al. [45] have shown a significantly lower concentration of this cytokine in diabetic patients than in the control group. The presence of dysfunctional regulatory T lymphocytes may be associated with the observed IL-2 deficiency [46]. IL-2 may affect Treg cell function and improve disease prognosis. The rejection of the proliferation signal for Treg from IL-2 accelerated disease progression in mice [47]. The use of antibodies to the IL-2 receptor resulted in an more frequent incidence of diabetes in mice [48]. In contrast, IL-2 therapy has contributed to Treg activation [49]. IL-10 and IL-4 demonstrate anti-inflammatory properties that affect the function and viability of pancreatic cells. Cytokines may show protective properties and protect against cell damage after cytotoxic reactions [50]. Kamiński et al. [51] showed that IL-4 modulates the synthesis of nitric oxides in macrophages, leading to pro-inflammatory cytokines’ contribution to the death of pancreatic beta cells. In some cell types, IL-4 has the ability to induce the synthesis of an antagonist of the IL-1β receptor, which reduces its activation upon binding to the receptor [52]. IL-4 exhibits properties that improve glucose tolerance and insulin effectiveness, and enhances glucose utilization. The enormous role of IL-4 may suggest that deficiency of this cytokine leads to metabolic disorders in patients with type 1 diabetes [53,54].

## 4. Materials and Methods

### 4.1. Patients and Control

The study group included 53 children with type 1 diabetes who were patients of the Department of Pediatric Endocrinology and Diabetology of the Medical University of Lublin in the period from October 2016 to February 2018. Patients diagnosed with type 1 diabetes based on the clinical picture and laboratory exponents of hyperglycemia, in accordance with the criteria of the Polish Diabetes Society.

The study group was divided into patients with newly diagnosed type 1 diabetes—group I (diagnosed type 1 diabetes within 3 months of inclusion in the study, after previous recovery from ketosis) and chronic patients—group II (duration of the disease over 3 months). 31 patients were qualified to group I, including 15 girls (48.4%) and 16 boys (51.6%). The studies were carried out in children aged 10.27 years (4–17 years) on average. Group II comprised 22 people, including 9 girls (40.91%) and 13 boys (59.09%). The average age of the respondents was 14.06 years (3.33 to 17.92 years).

The control group consisted of 20 children with an average age of 12.65 years (5.75–17.33 years)—5 girls (25%) and 15 boys (75%), diagnosed due to short stature at the Department of Pediatric Endocrinology and Diabetology at the Medical University of Lublin and the Endocrinology Clinic of the University Children’s Hospital in Lublin, in which physical examination and laboratory tests excluded glucose metabolism disorders, autoimmune diseases as well as somatotropinic and multi-hormonal pituitary insufficiency.

The study design was approved by the Bioethics Committee at the Medical University of Lublin (consent number KE-0254/299/2016). The funds for the study were obtained as part of DS 417. The study was conducted in accordance with the Helsinki Declaration.

Determination of glycemic control parameters: glycated hemoglobin (HbA1c), fructosamine, C-peptide concentration and peripheral blood counts were performed according to standard procedures of the ALAB Medical Analysis Laboratory at the University Children’s Hospital in Lublin. In the group of children with type 1 diabetes lasting more than 3 months, C-peptide levels were assessed retrospectively at the diagnosis of the disease.

### 4.2. Preparation of the Material

Peripheral blood was collected from the ulnar vein of patients and controls using sterile, EDTA-coated blood collection tubes (S-Monovette, SARSTEDT, Aktiengesellschaft and Co., Numbrecht, Germany) during hospitalization. The collected material was immediately used to assess the immunophenotype of lymphocytes, with particular emphasis on TLR2, TLR4, and TLR9 receptors, and to obtain plasma to assess the concentration of selected cytokines.

### 4.3. Assessment of Peripheral Blood Lymphocyte Subpopulation by Flow Cytometry

In order to assess the presence of lymphocyte surface antigens, the appropriate monoclonal antibodies were added to the test tubes in the amount of 20 µL per tube: anti-CD3/CD19-FITC/PE, anti-CD4/CD8/CD3-FITC/PE/PE-Cy5, anti-45/14-FITC/PE, anti-CD3/CD16+CD56-FITC/PE, anti-CD19-FITC, anti-CD4-FITC and anti-CD8-FITC from BD Biosciences (USA). Percentages of natural killer (NK) and natural killer T-like (NKT-like) cells were evaluated with flow cytometry using monoclonal antibody anti-CD3 FITC/CD16^+^CD56 PE (BD Biosciences, East Rutherford, NJ, USA), which allowed for simultaneous assessment of T CD3^+^ lymphocytes and NK cells CD3^−^/CD16^+^CD56^+^. Then, 50 µL of whole blood was added to each tube, followed by a 20-min incubation at room temperature. After incubation, 2 mL of lysis solution was added to each tube, and an 8-min incubation at room temperature was performed again. After this time, cells were washed twice with PBS buffered saline (700× *g*, for 5 min) and immediately analyzed on a FacsCalibur flow cytometer (Becton Dickinson, USA). The assessment of intracellular antigens was performed using the appropriate monoclonal antibodies: anti-TLR2-PE, anti-TLR4-PE, anti-TLR9-PE from BD Biosciences (USA) and the BD Cytofix/Cytoperm™ reagent kit (Becton Dickinson, USA) according to manufacturer’s protocol.

Data acquisition was performed using the FACS Diva Software 6.1.3 software, collecting 30,000 cells for each assay, while analyzing them using the CellQuest Pro program (Becton Dickinson, San Diego, CA, USA). The results of the cytometric analysis were presented as a percentage of cells stained with monoclonal antibodies conjugated with fluorescent dyes. Background fluorescence was determined, and the samples were gated to remove the cellular debris and include only single cells. An example of the cytometric analysis of T CD4+ lymphocytes, T CD8+ lymphocytes and B CD19+ lymphocytes expressing TLR-2 or TLR-4 is shown in the figure below (Figure 10A–C).

### 4.4. Assessment of the Concentration of IFN-γ, IL-2, IL-4 and IL-10 in the Plasma

Plasma cytokine concentration was assessed using the following ELISA kits (as recommended by the kit manufacturer): Human IFN-γ ELISA Kit, sensitivity < 8 pg/mL (R&D Systems, Minneapolis, MN, USA), Human IL-2 ELISA Kit, sensitivity < 7 pg/mL (R&D Systems, USA), Human IL-4 HS ELISA Kit, sensitivity 0.03–0.22 pg/mL (R&D Systems, USA) and Human IL-10 HS ELISA Kit, sensitivity 0.03–0.17 pg/mL (R&D Systems, USA).

The light absorbance readings in the tested samples were performed using an automated VICTOR reader (Perkin Elmer, Waltham, MA, USA). The concentration of the samples was assessed on the basis of a standard curve made of standards with different concentrations and plotted by the WorkOut computer program.

### 4.5. Statistical Analysis

The obtained data were statistically analyzed using the Statistica 12 PL computer program (StatSoft, Tulsa, OK, USA), assuming *p* < 0.05 as the level of significance. The normality of the distribution of continuous variables was assessed using the Shapiro-Wilk test. Statistical characteristics of continuous variables were presented in the form of medians, arithmetic means, standard deviations and extreme values (minimum and maximum). Intergroup comparisons for independent variables were performed using Student’s t tests for variables showing a normal distribution and Mann-Whitney U test for variables that did not show a normal distribution. Spearman’s rank correlation coefficient (R) was used to assess the strength and direction of the relationship between pairs of variables. The scatterplots were prepared with the use of Prism 8.4.2 computer program (Graphpad Software, San Diego, CA, USA).

## 5. Conclusions

The increase of lymphocytes expressing TLR2 and TLR4 receptors in patients’ cases with type 1 diabetes in the early stage of the disease and those treated chronically suggests that stimulation of these receptors accompanies the development of the disease.

The decrease of the number of CD8+ T and CD19+ B cells expressing the TLR9 receptor in patients’ cases at an early stage of the disease compared to patients treated chronically and demonstrated significant relationships with HbA1c and the concentration of fructosamine confirms that changes in the expression of this receptor may be involved in the regulation of glucose metabolism of patients with type 1 diabetes.

Considerably a higher number of CD19+ B lymphocytes, CD3+CD4+ T and CD3+CD8+ T lymphocytes and a lower number of NK cells of patients with newly diagnosed type 1 diabetes in the absence of significant changes in the number of these cells in chronic patients’ cases confirm the thesis of participation of these cells in early stage of the disease, which may contribute to the destruction of pancreatic beta cells.

A significantly higher concentration of IL-10 in the plasma of patients with newly diagnosed diabetes compared to patients treated chronically and the control group suggests an attempt to activate regulatory mechanisms in patients’ cases at an early stage of the disease, which do not constitute protection against the progressive progression of the disease.

There is an undeniable need for broader studies in order to better understand the immunological mechanisms leading to the development of type I diabetes. Such knowledge would be crucial in finding a potential agent that would interfere with that process slowing down or stopping the progression of the disease.

## Figures and Tables

**Figure 1 ijms-22-12135-f001:**
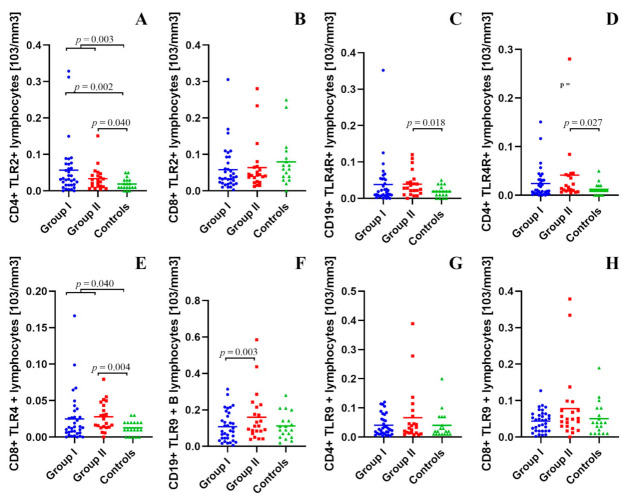
(**A**–**H**). Assessment of B CD19+, T CD4+, and T CD8+ lymphocytes with TLR2, TLR4, TLR9 expression in the study and control groups.

**Figure 2 ijms-22-12135-f002:**
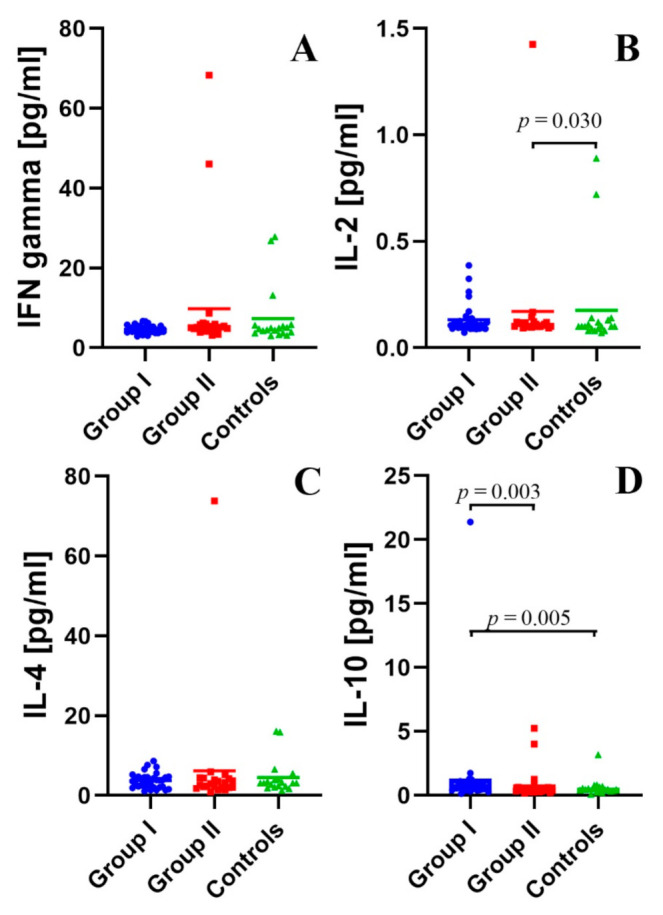
(**A**–**D**). Selected plasma cytokine levels.

**Figure 3 ijms-22-12135-f003:**
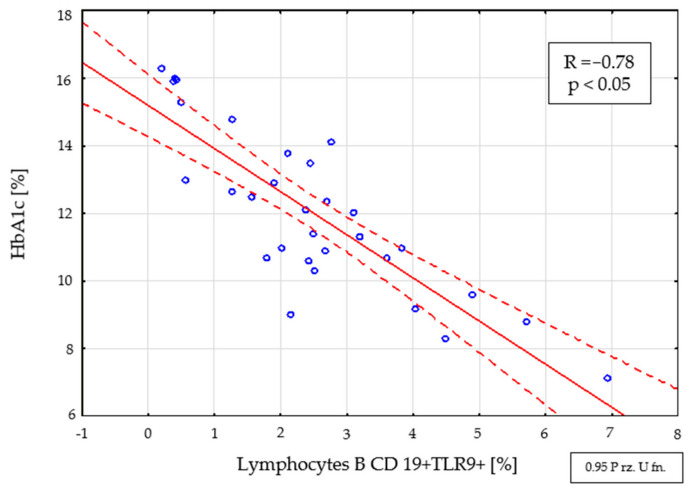
Relationship between the percentage of CD19+ TLR9+ B cells and the level of HbA1c in the group with newly diagnosed type 1 diabetes (<3 months from diagnosis).

**Figure 4 ijms-22-12135-f004:**
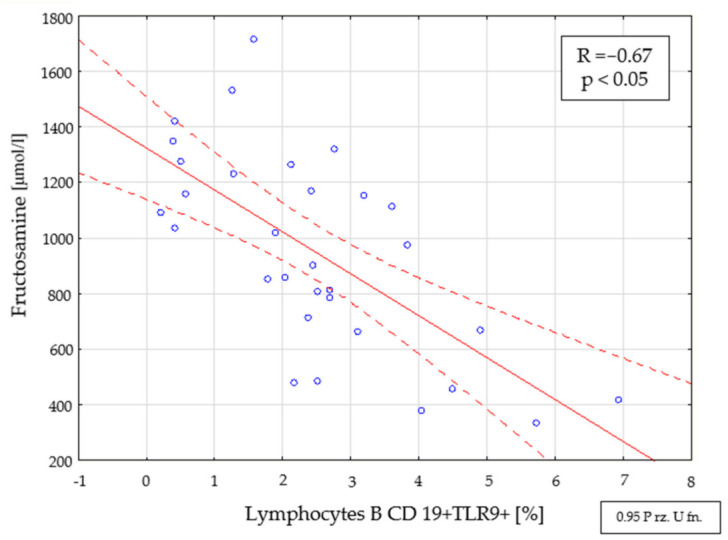
Relationship between the percentage of CD19+ TLR9+ B cells and the level of fructosamine in the group with newly diagnosed type 1 diabetes (<3 months from diagnosis).

**Figure 5 ijms-22-12135-f005:**
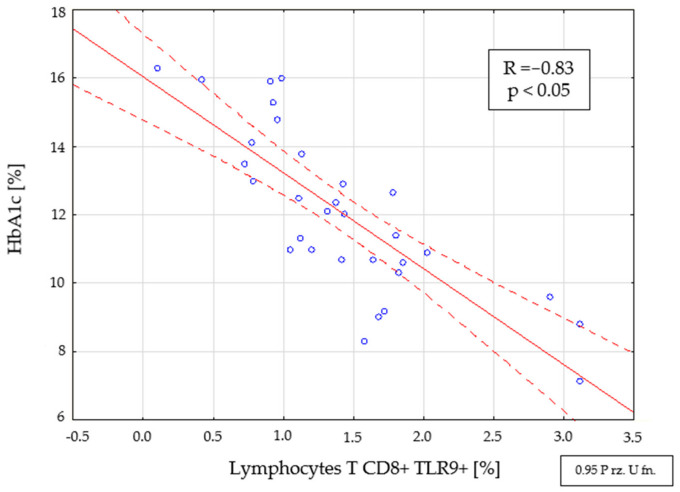
Relationship between the percentage of CD8+ TLR9+ T cells and the level of HbA1c in the group with newly diagnosed type 1 diabetes (<3 months from diagnosis).

**Figure 6 ijms-22-12135-f006:**
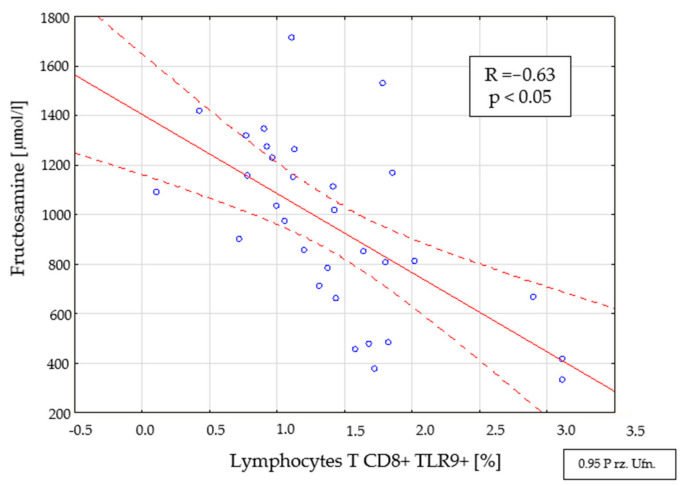
Relationship between the percentage of CD8+ TLR9+ T cells and the level of fructosamine in the group with newly diagnosed type 1 diabetes (<3 months from diagnosis).

**Figure 7 ijms-22-12135-f007:**
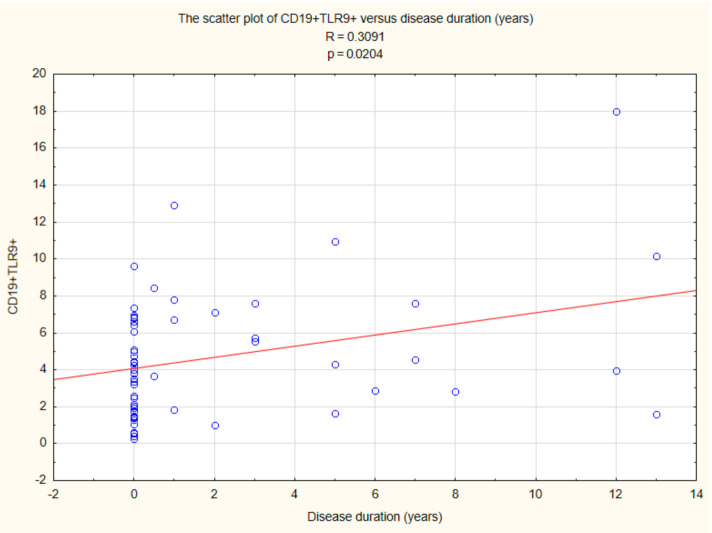
Relationship between CD19+ TLR9+ B cells and duration time after the onset of DM1.

**Figure 8 ijms-22-12135-f008:**
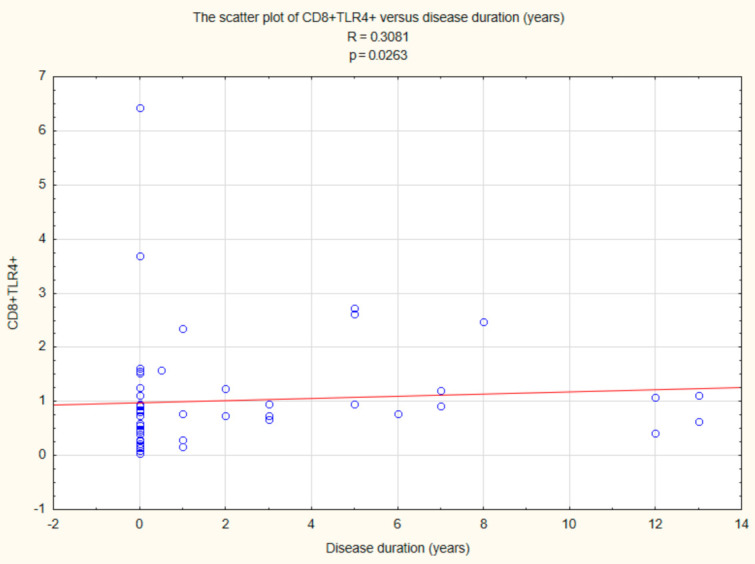
Relationship between CD8+ TLR4+ T cells and duration time after the onset of DM1.

**Figure 9 ijms-22-12135-f009:**
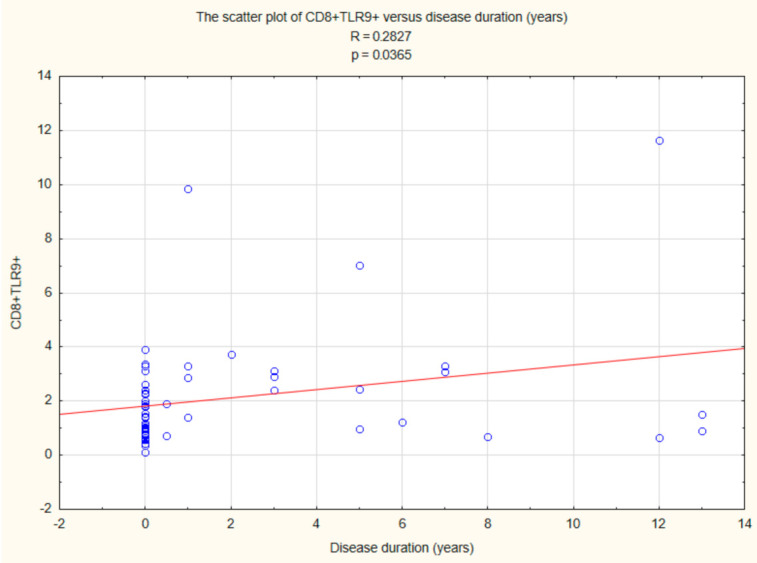
Relationship between CD8+ TLR9+ T cells and duration time after the onset of DM1.

**Figure 10 ijms-22-12135-f010:**
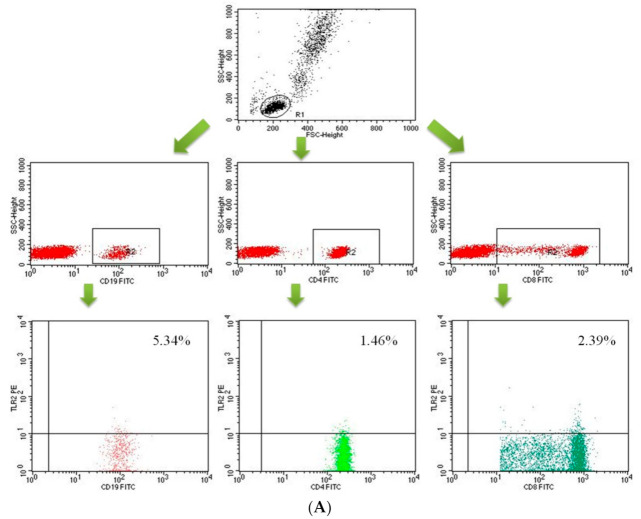
Dot-plots showing the percentage of B lymphocytes (CD19+) and T lymphocytes (CD4+, CD8+) with TLR2 expression (**A**), B lymphocytes (CD19+) and T lymphocytes (CD4+, CD8+) with TLR4 expression (**B**) B lymphocytes (CD19+) and T lymphocytes (CD4+, CD8+) with TLR9 expression (**C**).

**Table 1 ijms-22-12135-t001:** Basic characteristics of study and control groups.

Parameter		Mean	Median	Minimum	Maximum	SD
Glycated hemoglobin [%]	Group I	12	12.01	7.14	16	2.40
	Group II	8.8	9.20	5.70	16	2
Fructosamine [μmol/L]	Group I	950.04	974	333	1714	360.34
	Group II	513.10	520.50	286	7000	134.58
C-peptide [ng/mL]	Group I	0.46	0.34	0.10	1.89	0.39
	Group II	0.53	0.53	0.20	0.82	0.21
White blood cells [10^3^/μL]	Group I	9.63	8.40	4.18	24.32	4.28
	Group II	6.94	6.71	4.04	12.99	1.89
	Controls	6.43	5.91	4.05	8.99	1.38
Neutrocytes [10^3^/μL]	Group I	5.86	4.77	1.50	17.41	3.64
	Group II	3.55	3.35	1.60	6.90	1.22
	Controls	3.33	2.98	2.06	5.55	1.11
Lymphocytes [10^3^/μL]	Group I	2.88	2.80	0.72	4.53	0.95
	Group II	2.58	2.26	1.66	5.06	0.85
	Controls	2.33	2.53	1.23	3.48	0.67
Red blood cells [10^6^/μL]	Group I	5.06	4.84	4.14	6.68	0.59
	Group II	5.08	4.93	4.10	6.10	0.51
	Controls	4.79	4.79	4.04	5.28	0.34
Hemoglobin [g/dL]	Group I	14.04	14.20	11.60	16.20	1.25
	Group II	14.58	14.20	12.70	16.90	1.51
	Controls	13.66	13.85	11.70	16.00	0.97
Platelets [10^3^/μL]	Group I	274.39	262	99	534	82.35
	Group II	278.82	282.50	203	382	49.52
	Controls	291.69	289.50	221	400	50.05
Lymphocytes T CD3+ [%]	Study group	72.71	74.02	58.96	83.70	5.70
	Group I	73.00	74.02	58.96	83.70	6.01
	Group II	72.31	73.94	62.75	81.14	5.34
	Controls	68.13	70.94	52.19	75.22	6.89
Lymphocytes T CD3+ [10^3^/mm^3^]	Study group	2.017	1.851	0.537	3.860	0.740
	Group I	2.116	1.946	0.537	3.600	0.770
	Group II	1.879	1.598	1.148	3.860	0.670
	Controls	1.596	1.752	0.783	2.448	0.507
Lymphocytes B CD19+ [%]	Study group	14.61	14.28	4.64	30.76	5.23
	Group I	16.02	15.48	4.64	30.76	5.77
	Group II	12.63	12.79	6.43	20.53	3.64
	Controls	11.76	10.55	5.27	27.51	5.02
Lymphocytes B CD19+ [10^3^/mm^3^]	Study group	0.401	0.333	0.118	1.050	0.200
	Group I	0.454	0.461	0.118	1.050	0.212
	Group II	0.327	0.309	0.126	0.740	0.150
	Controls	0.271	0.252	0.092	0.527	0.121
Lymphocytes T CD3+CD4+ [%]	Study group	41.09	42.27	28.83	53.07	5.73
	Group I	40.91	42.41	28.83	53.07	6.27
	Group II	41.33	41.64	30.31	48.93	5.03
	Controls	37.37	36.07	28.56	51.68	7.35
Lymphocytes T CD3+CD4+ [10^3^/mm^3^]	Study group	1.131	1.043	0.382	2.090	0.420
	Group I	1.182	1.139	0.382	2.090	0.459
	Group II	1.063	0.947	0.667	2.080	0.360
	Controls	0.877	0.906	0.359	1.387	0.310
Lymphocytes T CD3+CD8+ [%]	Study group	27.30	27.91	15.30	42.06	4.93
	Group I	28.00	28.56	17.18	42.06	5.21
	Group II	26.35	26.81	15.30	32.92	4.47
	Controls	24.97	23.60	19.06	34.75	4.43
Lymphocytes T CD3+CD8+ [10^3^/mm^3^]	Study group	0.767	0.684	0.143	1.470	0.320
	Group I	0.825	0.827	0.143	1.470	0.340
	Group II	0.688	0.640	0.372	1.350	0.270
	Controls	0.587	0.581	0.285	0.998	0.212
CD3+CD4+/CD3+CD8+ ratio	Study group	1.57	1.46	0.80	3.10	0.46
	Group I	1.53	1.45	0.80	2.67	0.45
	Group II	1.63	1.52	0.97	3.10	0.46
	Controls	1.56	1.49	0.90	2.56	0.48
NK cells [%]	Study group	9.23	8.45	2.25	22.21	4.99
	Group I	7.43	7.06	2.25	14.07	3.49
	Group II	11.77	11.25	2.68	22.21	5.72
	Controls	13.34	12.56	4.05	22.62	5.47
NK cells [10^3^/mm^3^]	Study group	0.248	0.230	0.021	0.720	0.140
	Group I	0.214	0.188	0.021	0.570	0.124
	Group II	0.295	0.256	0.056	0.720	0.160
	Controls	0.309	0.247	0.110	0.654	0.165
The level of significance of differences between the studied groups in the percentage and number of parameter.						
**Parameter**		**Study Group/Controls**		**Group I/Controls**	**Group II/Controls**	**Group I/Group II**
Lymphocytes T CD3+ [%]		**0.009**		**0.016**	**0.042**	NS
Lymphocytes T CD3+ [10^3^/mm^3^]		**0.036**		**0.019**	NS	NS
Lymphocytes B CD19+ [%]		NS		**0.016**	NS	**0.019**
Lymphocytes B CD19+ [10^3^/mm^3^]		**0.015**		**0.003**	NS	**0.019**
Lymphocytes T CD3+CD4+ [%]		**0.038**		NS	NS	NS
Lymphocytes T CD3+CD4+ [10^3^/mm^3^]		**0.028**		**0.022**	NS	NS
Lymphocytes T CD3+CD8+ [%]		NS		**0.015**	NS	NS
Lymphocytes T CD3+CD8+ [10^3^/mm^3^]		**0.038**		**0.015**	NS	NS
CD3+CD4+/CD3+CD8+ ratio		NS		NS	NS	NS
NK cells [%]		**0.007**		**<0.001**	NS	**0.001**
NK cells [10^3^/mm^3^]		NS		**0.035**	NS	**0.041**

(NS—not significant).

## Data Availability

Due to privacy and ethical concerns, the data that support the findings of this study are available on request from [M.K.].

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
