# Peer review of "Impaired Innate Immunity in Pediatric Patients Type 1 Diabetes—Focus on Toll-like Receptors Expression"

_ijms, 2021, doi:10.3390/ijms222212135_

Round 1

Reviewer 1 Report

This report focused on subsets of the peripheral lymphocytes and the TLRs expression in Type 1 diabetes patients.   As a result, they showed significant differences in the level of TLRs expression and particular subsets of lymphocytes in newly diagnosed patients.  The authors speculated these phenomenons might reflect these cells killing beta-pancreas. 

There are several critical points as follows.

Major points

1. The differences in lymphocyte subsets and TLRs expression in DM1 might be due to hyperglycemia and metabolic disorder at the onset of the disease but not related to the beta cell destruction.

   How do the authors explain the relationships between the presented results and the beta cell destruction? 

2.The authors compared the data of newly diagnosed DM1 (<3 months from diagnosis) with that of patients with chronic DM1(> 3 months from diagnosis) to see the immune abnormality which was affected by the onset of DM1.   If the authors tried to show the immunoreaction related to the onset of DM1,they should show and analyze the relationships between the data abnormality and duration time after the onset of DM1.

Minor points

Figure 1-4   HbA1c %  the presented data should be x 100.

                   Fructosamine should be presented by μmol/l.

Author Response

Dear Reviewer 1,

thank you for your in-depth comments. We have corrected the manuscript as instructed. The article was proofread and corrected language- and style wise.

Point 1:

The differences in lymphocyte subsets and TLRs expression in DM1 might be due to hyperglycemia and metabolic disorder at the onset of the disease but not related to the beta cell destruction.

How do the authors explain the relationships between the presented results and the beta cell destruction? 

Response 1: As our reports are preliminary, a detailed analysis, also in the animal model, is advisable and we plan to expand our research.

Point 2:

The authors compared the data of newly diagnosed DM1 (<3 months from diagnosis) with that of patients with chronic DM1(> 3 months from diagnosis) to see the immune abnormality which was affected by the onset of DM1. If the authors tried to show the immunoreaction related to the onset of DM1, they should show and analyze the relationships between the data abnormality and duration time after the onset of DM1.

Response 2:  We performed additional correlation analyzes between the obtained values of CD19+ B cells, CD4+ and  CD8+ T cells with TLR2, TLR4, TLR9 expression and duration time after the onset of DM1 as recommended.

Point 3:

Figure 1-4   HbA1c%  the presented data should be x 100.

                   Fructosamine should be presented by μmol/l.

Response 3: We corrected minor mistakes in figures' legends as pointed out.

We hope, that you will find the revised article better and suitable for acceptance. In case there is something we have missed, please do not hesitate to tell us and we will implement the needed changes.

With best regards,

Paulina NiedĹşwiedzka-Rystwej

Reviewer 2 Report

Abstract:

The additional part of your study, which regards lymphocytes and IL-10 secretion, is already known from literature. Indeed the correlation between CD4+ and CD8+ in pediatric patients was already published in 2008; B cell are studied in details in this context; IL-10 secretion was evaluated in pediatric patients in 2008.

Introduction:

You should summarize from line 29 to line 53.

Results: 

You should replace tables with graphs, plotting the three different groups for each cell type of interested.

Materials:

It is not clear how you have identified NK cells: which antibody panel did you use to identify that cell type?

Bibliography:

It must be update.

Author Response

Dear Reviewer 2,

thank you for your in-depth comments. We have corrected the manuscript as instructed. The article was proofread and corrected language- and style wise.

Point 1:

Abstract: The additional part of your study, which regards lymphocytes and IL-10 secretion, is already known from literature. Indeed the correlation between CD4+ and CD8+ in pediatric patients was already published in 2008; B cell are studied in details in this context; IL-10 secretion was evaluated in pediatric patients in 2008.

Response 1: Abstract was corrected so that it does not appear as if we were the first team to study this subject.

Point 2:

Introduction: You should summarize from line 29 to line 53.

Response 2: Introduction lines 29 to 53 were summarized.

Point 3:

Results:  You should replace tables with graphs, plotting the three different groups for each cell type of interested.

Response 3: As hinted, we replaced the tables with scatterplots with applied statistical significance between the groups. The tables were moved to the supplementary materials section.

Point 4:

Materials: It is not clear how you have identified NK cells: which antibody panel did you use to identify that cell type?

Response 4: Percentages of natural killer (NK) and natural killer T-like (NKT-like) cells were evaluated with flow cytometry using monoclonal antibody anti-CD3 FITC/CD16+CD56 PE (BD Biosciences, USA), which allowed for simultaneous assessment of T CD3+ lymphocytes and NK cells CD3−/CD16+CD56+.

Point 5:

Bibliography: It must be update.

Response 5: We revised the bibliography and did not find any references that needed to be updated, as they are in compliance with the citing instructions for the authors of Int. J. Mol. Sci.

We hope, that you will find the revised article better and suitable for acceptance. In case there is something we have missed, please do not hesitate to tell us and we will implement the needed changes.

With best regards,

Paulina NiedĹşwiedzka-Rystwej

Round 2

Reviewer 2 Report

Dear Authors,

 the revised version of manuscript is fine. It can be accepted in the present form.

 Regards